# Continuous Production of Functionalized Graphene Inks by Soft Solution Processing

**DOI:** 10.3390/nano13142043

**Published:** 2023-07-11

**Authors:** Kodepelly Sanjeeva Rao, Jaganathan Senthilnathan, Jyh-Ming Ting, Masahiro Yoshimura

**Affiliations:** 1Promotion Center for Global Materials Research (PCGMR), Department of Material Science and Engineering, National Cheng Kung University, Tainan 701, Taiwan; 2Department of Civil Engineering, Indian Institute of Technology Madras (IIT Madras), Chennai 600036, Tamil Nadu, India; 3Hierarchical Green-Energy Materials (Hi-GEM) Research Center, National Cheng Kung University, Tainan 701, Taiwan

**Keywords:** solution processing, functionalized graphene, continuous production, inks

## Abstract

The continuous production of high-quality, few-layer graphene nanosheets (GNSs) functionalized with nitrogen-containing groups was achieved via a two-stage reaction method. The initial stage produces few-layer GNSs by utilizing our recently developed glycine-bisulfate ionic complex-assisted electrochemical exfoliation of graphite. The second stage, developed here, uses a radical initiator and nitrogen precursor (azobisisobutyronitrile) under microwave conditions in an aqueous solution for the efficient nitrogen functionalization of the initially formed GNSs. These nitrile radical reactions have great advantages in green chemistry and soft processing. Raman spectra confirm the insertion of nitrogen functional groups into nitrogen-functionalized graphene (N-FG), whose disorder is higher than that of GNSs. X-ray photoelectron spectra confirm the insertion of edge/surface nitrogen functional groups. The insertion of nitrogen functional groups is further confirmed by the enhanced dispersibility of N-FG in dimethyl formamide, ethylene glycol, acetonitrile, and water. Indeed, after the synthesis of N-FG in solution, it is possible to disperse N-FG in these liquid dispersants just by a simple washing–centrifugation separation–dispersion sequence. Therefore, without any drying, milling, and redispersion into liquid again, we can produce N-FG ink with only solution processing. Thus, the present work demonstrates the ‘continuous solution processing’ of N-FG inks without complicated post-processing conditions. Furthermore, the formation mechanism of N-FG is presented.

## 1. Introduction

Graphene-based materials, particularly nitrogen-functionalized graphene (N-FG) [1,2,3,4,5,6,7,8,9,10,11,12,13,14,15,16,17], have attracted attention in the synthesis community due to their potential applications in the fields of catalysis, energy storage, electronics, sensors, and biomedical applications [18,19,20,21,22,23]. For example, N-FG can behave as an electron-rich n-type field-effect transistor [17] or as a metal-free catalyst [12] in the oxygen reduction reaction in alkaline fuel cells, batteries, and ultracapacitors. A variety of synthetic methods for N-FG, such as pyrolysis [1], plasma processing [2], chemical vapor deposition [5], and solution processing [11,12,13,14,15,16,24,25,26,27], have thus been developed. Among these methods, solution processing is very important due to its low cost and environmental friendliness. Interestingly, functionalized graphene production using electrochemical approaches is still rare [27,28,29]. Parvez et al. [1] produced N-FG with 2.04% to 19.46% nitrogen content via a multi-stage, high-temperature pyrolysis method using a high quantity of nitrogen precursor. Interestingly, a solvothermal approach could achieve up to 16.4% nitrogen content in N-FG [30]. However, multi-step nitrogen functionalization methods are rather complex. The development of simple and efficient synthetic methods for the preparation of N-FGs is thus important. In this context, we have been developing electrochemical exfoliation methods [29,31], particularly for the continuous production of N-FG [27]. Nevertheless, the recent report by Ustavytska et al. [32] on the one-step electrochemical method for the preparation of nitrogen-doped graphene is highly promising.

The heteroatom-doping and introduction of functional groups in GNSs have been studied in the literature for some time [33,34,35,36,37,38,39,40,41,42]. Yadav et al. reviewed the synthesis and characterization of N-doped Graphene [43]. Recent advances in N-doped graphene for potential application in energy storage devices such as rechargeable batteries have been reviewed by Ikram et al. [44]. Xu et al. reviewed the synthesis and characterization of nitrogen-doped graphene articles for energy-related applications [45]. The development of chemical vapor deposition methods for the preparation of nitrogen-doped graphene and application in the area of field effect transistors and energy storage devices has been reviewed by Deokar et al. [46]. Nevertheless, Sreeprasad et al. reviewed the effect of functionalization on the electrical properties of graphene [47]. Therefore, doped and/or functionalized GNSs have been a very promising field of research due to the possibility of designing and developing novel materials with excellent properties for a wide range of applications. The major drawbacks of many synthesis methods for nitrogen-doped or functionalized graphene include multiple steps, long duration, the use of expensive equipment, harsh processing conditions, and environmentally toxic chemicals. Unlike these conventional methods, the proposed continuous production approach has several advantages, such as simple and environmentally friendly soft-processing, which can be economical, faster, and time- and energy-saving. Furthermore, for the patterning of functional materials by printing followed by sintering, particles usually need to be dispersed in a liquid along with other additives to produce the so-called ‘inks’ Interestingly, the proposed continuous method produces stable dispersions of N-FG in several solvents, indicating the possibility of the development of N-FG inks for various applications through direct printing/patterning, where sintering may not be required [48,49].

As an advancement of our continuous production approach [27], this study proposes a novel synthetic approach for the continuous production of N-FG nanosheets directly from graphite via a two-stage reaction. The initial stage produces high-quality, few-layer graphene nanosheets (GNSs, product I) in an aqueous solution via our recently developed exfoliation approach [29] that involves the glycine-bisulfate ionic complex-assisted electrochemical exfoliation of graphite. In the second stage, the direct nitrogen functionalization of the initially formed GNSs is conducted in an aqueous solution using azobisisobutyronitrile (AIBN, [(CH_3_)_2_C(CN)]_2_N_2_) as a simple nitrogen precursor under microwave irradiation at ambient conditions. Under these conditions, N-FG (product II) is produced directly from graphite. This method does not use any additional processing, such as filtration, drying, or firing processes, unlike complex nitrogen functionalization methods reported in the literature [1,18,19]. Therefore, this solution-based method is proposed as a continuous soft process for the direct formation of N-FG nanosheets starting from abundantly available graphite. N-FG nanosheets are very important in various applications, such as catalysis [23,27], as they disperse in various solvents well due to the insertion of active nitrogen centers. Furthermore, we would like to propose that inks of N-FG can be produced by this processing method continuously within a solution [27,29,31]. Nevertheless, the characterization of N-FG using various techniques was conducted, and the formation mechanism was presented in this article. Future research may involve further development of continuous soft solution processing of a wide range of materials (for example, organic–inorganic hybrid materials) and optimization of customized inks.

## 2. Experimental Section

### 2.1. Experimental Setup

Electrochemical experiments were performed according to our developed method [29]. All chemicals were purchased from Sigma-Aldrich and used directly. High-pressure liquid chromatography (HPLC)-grade solvents were used. Milli-Q water (>18.2 MΩ) was obtained from a Roda purification system (Te Chen Co., Ltd., Taichung City, Taiwan). The initially formed GNS [29] reaction mixture was basified to pH 10 by gradually adding aqueous 1.0 M NaOH at 0 °C and AIBN (25 mg). The reaction mixture was then subjected to microwave irradiation (90 W, 160 °C, 180 psi, 5 min). The resulting aqueous dispersion was cooled to room temperature, neutralized with 1.0 M HCl solution, filtered with 100 nm porous filters, washed with deionized water by vacuum filtration, and redispersed into suitable solvents using ultrasonication for 5 min. The supernatant dispersions were used for further characterization.

### 2.2. Characterizations

High-resolution transmission electron microscopy (HR-TEM, JEOL, JSM 2100F) was used to monitor the micro/nanostructure and surface morphology of N-FG at 200 kV. High-resolution X-ray photoelectron spectroscopy (XPS) measurements (PHI Quantera SXM ULVAC Inc., Kanagawa, Japan) were used to determine the carbon and nitrogen binding energies of N-FG. Raman spectral measurements of graphite and N-FG were performed using a confocal micro-Raman spectrometer (Renishaw inVia) at an excitation wavelength of 633 nm. The Si peak at 520 cm^−1^ was used as a reference. Fourier transform-infrared (FT-IR) spectra were recorded using a Vertex 70 FT-IR spectrometer (Bruker, Karlsruhe, Germany). Ultraviolet-visible (UV-Vis) spectra were recorded using a spectrophotometer (JASCO V-670). ^1^H NMR (nuclear magnetic resonance) spectra were recorded using a Bruker DRX600 high-resolution NMR spectrometer. Redispersed solution was drop-casted on lacy carbon-coated Cu grids to prepare TEM samples. Redispersed materials were deposited on a glass substrate and evaporated (60 °C, 30 min) to prepare Raman spectroscopy and XPS samples. UV-Vis samples were prepared from equal dilutions of all materials with milli-Q water. FT-IR samples were prepared using the KBr disc method.

## 3. Results and Discussion

### 3.1. The Formation of N-FG

The experimental schematic of the N-FG formation process is shown in Figure 1a. In the initial stage, high-quality GNSs are prepared using our recently developed [29] electrochemical strategy that utilizes 15 wt% glycine-bisulfate and 85% water solution with working bias voltages of +1 V for 5 min and +3 V for 5 min at room temperature and atmospheric pressure. The resulting reaction mixture is basified with aqueous NaOH solution under microwave irradiation [50]. Figure 1a also shows photographs of exfoliated GNSs (product I) and N-FG (product II) nanosheets in the electrolyte solution. Furthermore, N-FG dispersions in dimethyl formamide (DMF), ethylene glycol (EG), acetonitrile (ACN), and water were prepared. These dispersions were stable for up to a month (Figure 1b). It clearly demonstrates that those dispersions can be used as inks for further printing on any substrates.

### 3.2. Structural Features of N-FG

Raman spectroscopy is a powerful tool for detecting defects and confirming nitrogen doping in nanomaterials. The Raman spectrum of N-FG (Figure 2) shows a D band (disorder mode) at 1331 cm^−1^, a D’ shoulder band (disorder of edge carbons) at 1618 cm^−1^, a G band (ordered in-plane sp^2^ carbon atoms) at 1578 cm^−1^, and a 2D band (generated from two-phonon double resonance) at 2680 cm^−1^. The Raman results of N-FG show good agreement with results published in our recent paper [51]. The D band to G band intensity ratio (I_D_/I_G_) [52,53] of N-FG is higher (0.82) than that of starting graphite (0.54), indicating the insertion of oxygen and nitrogen functional groups and partial disorder at carbon edges. The presence of nitrogen functional groups also indicates enhanced dispersibility of N-FG in DMF, EG, and ACN (Figure 1b) under 5 min sonication when compared to 10 min sonication of GNS dispersion [29]. These dispersions may be potentially used as functionalized graphene inks. Thus, the continuous production of high-quality N-FG nanosheets is achieved under mild conditions.

TEM observations of N-FG show the presence of high-quality, few-layer GNSs (Figure 3). Thin nanosheets appear interconnected in a low-magnification TEM image (Figure 3a), and only a few defects appear in the HR-TEM image (Figure 3b). Few-layer N-FG with a 0.35 nm lattice spacing can be observed (Figure 3c,d). The selected area electron diffraction (SAED) pattern of N-FG (Figure 3c, inset) shows few defects. The observed lattice spacing of N-FG is about 0.35 nm, which is in good agreement with a previous report [51]. The quantitative energy-dispersive X-ray spectroscopy (EDS) mapping of the HR-TEM image of an N-FG nanosheet (Figure 3e) shows a uniform distribution of carbon (Figure 3f), nitrogen (Figure 3g), and oxygen (Figure 3h) functional groups. Furthermore, since the 2D band appears at 2680 cm^−1^ in the Raman spectrum of N-FG, this confirms the presence of few-layer graphene [52,53].

Figure 4 shows XPS spectra of graphite and N-FG. The XPS results demonstrate the doping concentration and chemical states of doped nitrogen functional groups into N-FG. Nitrogen was additionally detected in the wide-scan XPS spectrum of N-FG when compared to that of graphite (Figure 4a). The C/O/N ratio of N-FG is 87:9:4, whereas the C/O ratio of graphite is 98.5:1.5, confirming the insertion of nitrogen functional groups into N-FG. The C1s core level spectra of graphite (Figure 4b) and N-FG (Figure 4c) confirm the presence of graphitic bonds (C-C) at 284.8 eV. The graphite C-OH bonds appeared at 285.8 eV, whereas the C-OH/C-N bonds of N-FG appeared at 285.6 eV. Additionally, N-FG showed a new C-O (alkoxy) bond at 286.5 eV. These results are in good agreement with those published in the literature [54,55,56]. The N1s core level spectra of N-FG (Figure 4d) can be deconvoluted into three bands, which are assigned to pyridinic nitrogen (399.3 eV), pyrrolic nitrogen (400.2 eV), and graphitic nitrogen (401.2 eV), in good agreement with reported results [57].

Figure 5 shows the FTIR spectrum of N-FG, which further confirms the insertion of nitrogen functional groups. Peaks in the region of 990–1430 cm^−1^ can usually be attributed to CH in-plane bending vibrations [58,59]. The bands at 993 and 1065 cm^−1^ are attributed to δ(C-H) bending vibration. Peaks attributed to the stretching vibration of methyl-substituted carbon–carbon double bonds γ(C=C) appear at 1420, 1469, and 1625 cm^−1^, and that at 1659 cm^−1^ is attributed to =CH vibrations [60,61]. The peaks at around 2859 and 2922 cm^−1^ are attributed to asymmetric and symmetric -CH stretching vibration modes, respectively. Peaks corresponding to aromatic C-H bending vibrations [60] appeared at around 1420 and 1469 cm^−1^. The bands at 2039, 2203, and 3446 cm^−1^ correspond to -CH=N and -CH-NH groups [61,62]. The presence of nitrogen functional groups in N-FG was further confirmed by significant dispersibility in various solvents.

We postulate that the surface modification of graphene materials via functionalization is a versatile method for tuning their chemical and electronic properties. The structural features and formation mechanism of N-FG are presented in Figure 6. The GNS structure is composed of hydroxyl/phenolic, epoxide, and carboxylic acid groups [29]. The treatment of GNSs with aqueous NaOH solution neutralized HSO_4_^−^ ions from Gly·HSO_4_. The reaction of GNS with AIBN at 160 °C and 180 psi under microwave irradiation formed N-FG. During microwave irradiation, AIBN undergoes the formation of isobutyronitrile radicals. These radicals react with functional groups and the aromatic network present in GNSs and subsequently form radicalized graphene. This radicalized graphene further reacts with nitrile radicals to form N-FG via the insertion of various kinds of nitrogen functional groups. The XPS spectrum of N-FG also confirms the insertion of 4% nitrogen. Thus, the formation of N-FG via radical-induced reactions of GNSs in aqueous NaOH solution is a green chemistry approach. Therefore, the proposed method is applicable not only to the continuous production of N-FG but also to the production of diverse nitrogen-functionalized materials. We expect that the solution processing method can be extended for continuous production of hybrid materials such as metal supported/N-FG, which may find potential application in the area of catalysis [63,64,65,66,67,68,69]. It is expected that these materials will have a wide range of surface area and pore size distribution, which can be useful for various applications such as catalysis and energy storage devices [70]. The bulk electrical conductivity of these N-FG materials may be comparable to parent graphene-based material, which can be optimized for the designing and development of electrodes [70,71]. Nevertheless, it has been demonstrated that more functionalized catalysts like Au/N-FG hybrids can be prepared in the subsequent step [27].

## 4. Conclusions

This study developed a simple two-stage potential method for the continuous production of high-quality, few-layer N-FG nanosheets via the Gly·HSO_4_ ionic complex-assisted electrochemical exfoliation of graphite and subsequent nitrogen functionalization under isobutyronitrile radical conditions. The advantages of this method include low cost, continuous production, fast processing, and possible suitability for mass production. Additionally, radical reactions in aqueous solutions reflect advantages like green chemistry and soft solution processing. The proposed formation mechanism of N-FG reveals that isobutyronitrile radicals derived from AIBN plays a key role in the insertion of edge/surface nitrogen functional groups. The stable dispersions of N-FG nanosheets in solvents such as DMF, EG, ACN, and H_2_O are useful for catalysis and other functional applications. In those applications, the stable dispersion of N-FG in solutions as inks is important because the stable inks can be printed on various substrates. Furthermore, the N-FG can be further processed continuously in solution for the preparation of functional hybrids with metals and metal oxides for a wide range of applications. Nevertheless, the simplicity of the proposed continuous soft processing approach makes the proposed method suitable for the production of diverse 2D materials suitable for various applications.

## Figures and Tables

**Figure 1 nanomaterials-13-02043-f001:**
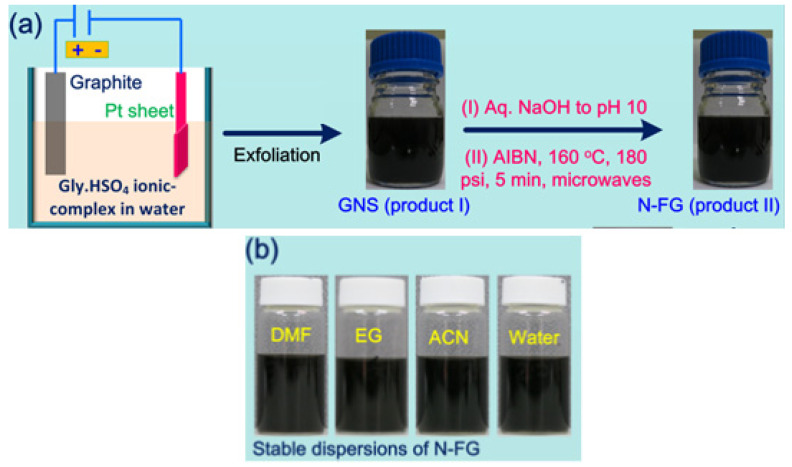
Schematic of N-FG formation process. (**a**) Electrochemical experimental setup diagram (left) and photographs of GNSs (product I) and N-FG (product II) directly in electrolyte; (**b**) Photographs of purified N-FG dispersions in DMF, EG, ACN, and water, respectively.

**Figure 2 nanomaterials-13-02043-f002:**
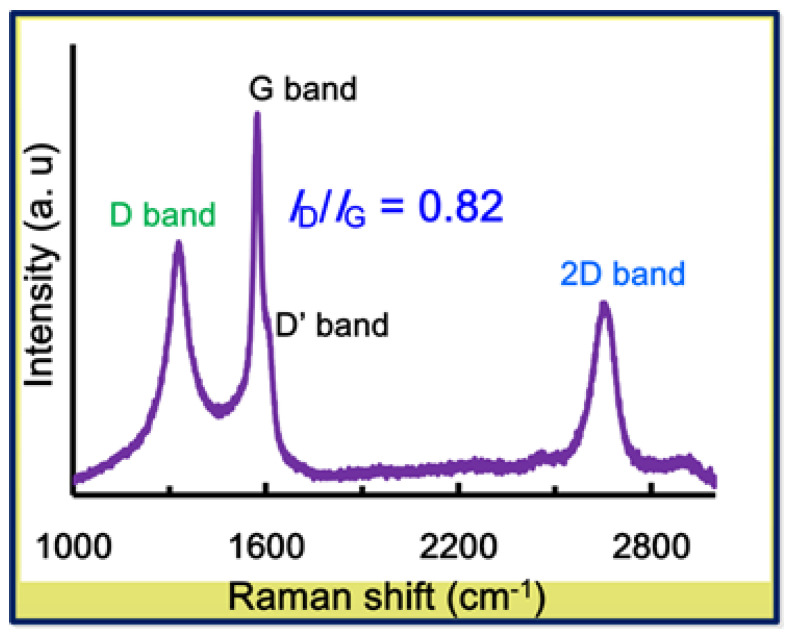
Raman spectrum of N-FG, indicating presence of high-quality, few-layer GNSs.

**Figure 3 nanomaterials-13-02043-f003:**
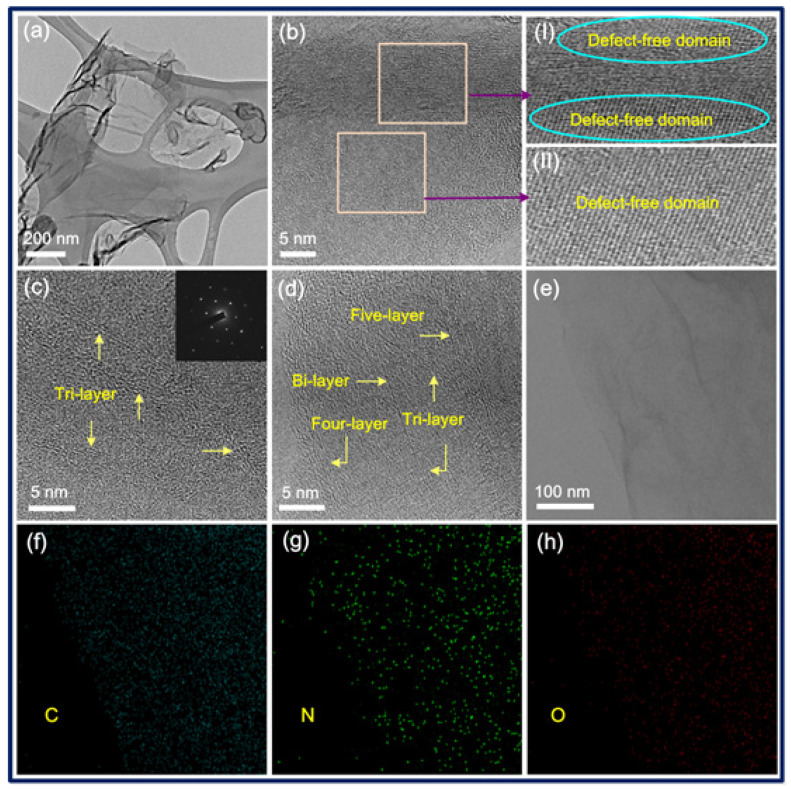
HR-TEM images of N-FG showing (**a**) interconnected thin sheets, (**b**) presence of few defects, and (**c**,**d**) few-layer nanosheets. Inset of (**c**) shows SAED patterns of N-FG. (**e**) TEM image of N-FG and elemental mapping of (**f**) C, (**g**) N, and (**h**) O of N-FG.

**Figure 4 nanomaterials-13-02043-f004:**
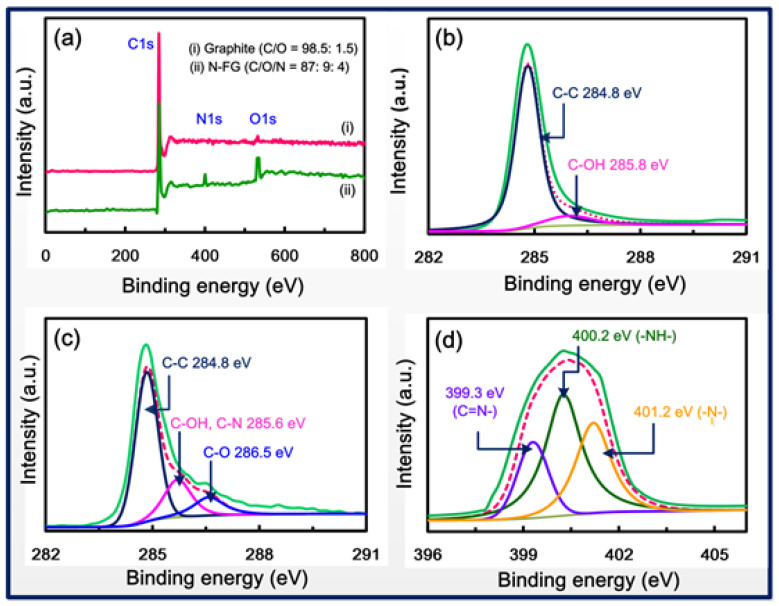
XPS measurements. (**a**) Wide-scan XPS spectra of (i) graphite and (ii) N-FG, C1s core level spectra of (**b**) graphite and (**c**) N-FG, and (**d**) N1s core level spectra of N-FG.

**Figure 5 nanomaterials-13-02043-f005:**
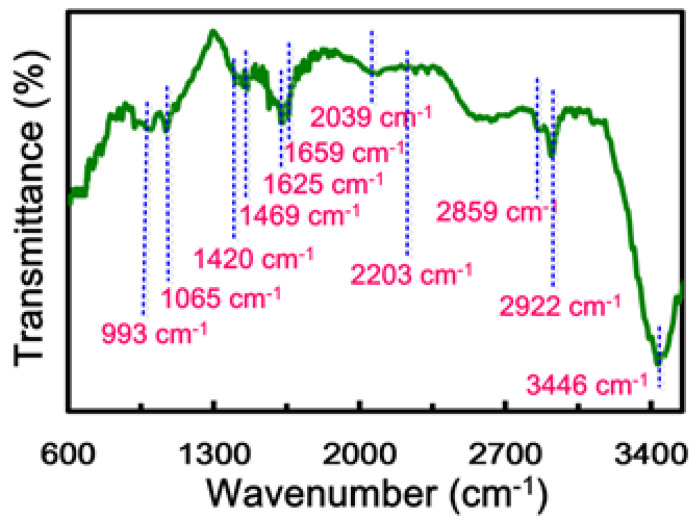
FT-IR spectrum of N-FG.

**Figure 6 nanomaterials-13-02043-f006:**
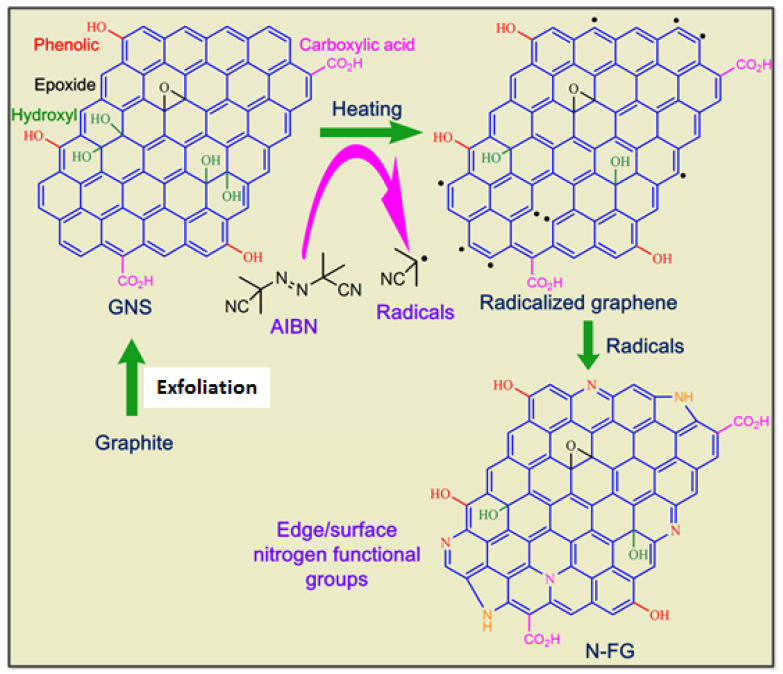
Schematic of proposed formation mechanism of N-FG.

## Data Availability

No new data available.

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
