# Peer review of "Continuous Production of Functionalized Graphene Inks by Soft Solution Processing"

_nanomaterials, 2023, doi:10.3390/nano13142043_

Round 1
Reviewer 1 Report
This is a paper presenting the preparation of N-functionalised graphene and its chemical and SEM/TEM characterisation. The paper is currently of no great interest to the readers. I would suggest the following improvements in product characterisation:
a. Find pore size distribution, specific surface area and specific surface volume of product (that could be of interest to energy storage R&D for example: https://doi.org/10.3390/nano11112899)
b. Measure electrical conductivity of your produced graphene.
Reviewer 2 Report
In their paper, the Authors introduce a method to obtain graphene-nanosheets and dope them with nitrogen in an efficient and continuous way, which could be particularly useful to obtain nitrogen-functionalized graphene nanosheet inks.
They describe the synthesis technique they have adopted (starting from the electrochemical exfoliation technique they have previously reported), they show the results of the material characterization (which proves nitrogen functionalization) and they present an explanation of the formation mechanism.
The paper is very interesting, well organized and written, and the content of the article is scientifically sound.
Therefore, I suggest its publication, with some minor corrections and improvements that I detail in the following.
a) Please improve the unclear sentences on line 40 ("even that could involve") and on line 108 ("aqueous min").
b) Is it possible that the obtained material presents also nitrogen doping, beyond functionalization with nitrogen groups, or the performed characterization excludes this possibility?
c) I suggest to add in the introduction some other references on graphene doping, such as
Wei, D.; Liu, Y.; Wang, Y.; Zhang, H.; Huang, L.; Yu, G. Synthesis of N-Doped Graphene by Chemical Vapor Deposition and Its Electrical Properties. Nano Lett. 2009, 9, 1752–1758; doi: 10.1021/nl803279t;
Marconcini, P.; Cresti, A.; Roche S. Effect of the Channel Length on the Transport Characteristics of Transistors Based on Boron-Doped Graphene Ribbons. Materials 2018, 11, 667; doi: 10.3390/ma11050667.
